# The correlation between cellular O-GlcNAcylation and sensitivity to O-GlcNAc inhibitor in colorectal cancer cells

**Pawaris Wongprayoon**[1,2], **Supusson Pengnam**[1,3,4], **Roongtiwa Srisuphan**[2], **Praneet Opanasopit**[3,4,5], **Siwanon Jirawatnotai**[6,7,8], **Purin Charoensuksai**[1,2]*

1 Faculty of Pharmacy, Department of Biomedicine and Health Informatics, Silpakorn University, Nakhon Pathom, Thailand, 2 Faculty of Pharmacy, Bioactives from Natural Resources Research Collaboration for Excellence in Pharmaceutical Sciences (BNEP), Silpakorn University, Nakhon Pathom, Thailand, 3 Faculty of Pharmacy, Center of Precision Medicine Innovation and Advanced Medicinal Product Development, Silpakorn University, Nakhon Pathom, Thailand, 4 Faculty of Pharmacy, Green Innovations Group (PDGIG), Silpakorn University, Nakhon Pathom, Thailand, 5 Faculty of Pharmacy, Department of Industrial Pharmacy, Silpakorn University, Nakhon Pathom, Thailand, 6 Faculty of Medicine Siriraj Hospital, Siriraj Center of Research Excellence for Precision Medicine and Systems Pharmacology, Mahidol University, Bangkok, Thailand, 7 Faculty of Medicine Siriraj Hospital, Department of Pharmacology, Mahidol University, Bangkok, Thailand, 8 Faculty of Pharmacy, Silpakorn University, Nakhon Pathom, Thailand

* charoensuksai_p@su.ac.th

**Data Availability Statement:** All relevant data are within the manuscript and its Supporting Information files.

## Abstract

The upregulation of O-GlcNAc signaling has long been implicated in the development and progression of numerous human malignancies, including colorectal cancer. In this study, we characterized eight colorectal cancer cell lines and one non-cancerous cell line for O-GlcNAc-related profiles such as the expression of OGT, OGA, and total protein O-GlcNAcylation, along with their sensitivity toward OSMI-1 (Os), an OGT inhibitor (OGTi). Indeed, Os dose-dependently suppressed the viability of all colorectal cancer cell lines tested. Among the three O-GlcNAc profiles, our results revealed that Os $IC_{50}$ exhibited the strongest correlation with total protein O-GlcNAcylation (Pearson Correlation Coefficient r = -0.73), suggesting that total O-GlcNAcylation likely serves as a better predictive marker for OGTi sensitivity than OGT expression levels. Furthermore, we demonstrated that Os exhibited a synergistic relationship with regorafenib (Re). We believed that this synergism could be explained, at least in part, by the observed Re-mediated increase of cellular O-GlcNAcylation, which was counteracted by Os. Finally, we showed that the Os:Re combination suppressed the growth of NCI-H508 tumor spheroids. Overall, our findings highlighted OGTi as a potential anticancer agent that could be used in combination with other molecules to enhance the efficacy while minimizing adverse effects, and identified total cellular O-GlcNAcylation as a potential predictive marker for OGTi sensitivity.

**Funding:** This report is part of the research project "The effect of OGT inhibitor in the four molecular subclasses of colorectal cancer" financially supported by the Research Grant for New Scholar (RGNS) from the Office of the Permanent Secretary, Ministry of Higher Education, Science, Research, and Innovation, Fiscal Year 2563 BE [Grant number RGNS 63-225], awarded to PC. There was no additional external funding received for this study. The funder had no role in study design, data collection and analysis, decision to publish, or preparation of the manuscript.

**Competing interests:** The authors have declared that no competing interests exist.

## 1. Introduction

Ranked third among the most prevalent types of cancer and cancer-associated death [1], colorectal cancer (CRC) remains a significant global health challenge [2]. The incidence of CRC has risen notably in Western nations, attributed in part to the aging population [3]. Despite advancements in diagnostic technologies for early detection and the approval of novel anticancer medications, little improvement has been observed in cure rates and long-term survival over recent decades [3]. Currently, treatment selection of colorectal cancer primarily relies on cancer staging and tumor location [4]. However, despite adherence to standard treatment protocols, clinical outcomes still vary significantly, underscoring the substantial gap between current treatment approaches and personalized CRC therapy [3,5].

Regorafenib (BAY 73–4506, Stivarga®, Bayer) is an oral multi-kinase inhibitor that has demonstrated efficacy in the CORRECT trial for the treatment of metastatic colorectal cancer (mCRC), particularly in patients refractory to standard therapies, including fluoropyrimidine, oxaliplatin, irinotecan, anti-vascular endothelial growth factor therapy, and anti-EGFR agents for those with wild-type KRAS [6]. The CORRECT trial evaluated regorafenib as a monotherapy compared to placebo with best supportive care. The results showed a favorable outcome for regorafenib over control group, including improved median overall survival (6.4 months vs. 5.0 months) and progression-free survival (1.9 months vs. 1.7 months), leading to its clinical approval by the US-FDA in 2012 and the EMA in 2013 [7]. Mechanism of action of regorafenib involves inhibiting multiple kinases involved in tumor growth and angiogenesis, including vascular endothelial growth factor receptor (VEGFR), tyrosine kinase with immunoglobulin and epidermal growth factor homology domain 2 (TIE2), fibroblast growth factor receptor (FGFR), platelet derived growth factor receptor (PDGFR), RET, KIT and RAF [8].

By inhibiting these critical pathways, regorafenib reduces tumor proliferation and disrupts angiogenesis. Its activity to target multiple oncogenic pathways highlights its potential in mCRC management and further research.

First discovered in 1984, O-GlcNAcylation (O-linked-β-N-acetylglucosaminidation) is a post-translational modification characterized by the attachment of N-Acetyl-D-glucosamine moiety to serine and threonine residues of target proteins [9]. O-GlcNAcylation is governed by two enzymes i.e. O-GlcNAc transferase (OGT) and O-GlcNAcase (OGA) which adds and removes O-GlcNAc moieties, respectively. Elevated OGT expression and increased cellular O-GlcNAcylation have been observed in many cancer types including prostate, breast and colorectal cancer (reviewed in [10]). Over the years, O-GlcNAcylation has been linked to multiple cancer hallmarks through the modification and modulation of many cancer-relevant proteins such as FOXM1 [11], c-MYC [12], β-catenin [13], snail [14] and NF-kB [15,16]. Furthermore, depletion of O-GlcNAc in cancer cells via si/shRNA has shown promising outcomes in cancer treatment including decreased proliferation, inhibited anchorage independent growth, suppression of metastatic phenotypes and sensitizing cancer cells to certain anticancer drugs [11–13,17,18]. Given the attractiveness of OGT as a potential molecular target for cancer treatment, several small molecule inhibitors of OGT (OGTi) have been developed [19–22].

Despite the evidence indicating the promise of targeting OGT for CRC treatment, the effects of OGT inhibition among CRC cell lines remain largely unexplored. In this study, the we characterized eight CRC cell lines for O-GlcNAc-related profiles including the expression of OGT, OGA, and total protein O-GlcNAcylation, along with their sensitivity toward an OGTi OSMI-1 (Os). We provided evidence that, among the three O-GlcNAc profiles, total protein O-GlcNAcylation emerged as a potential predictive marker OGTi sensitivity.

Furthermore, we demonstrated that combining Os with a well-known anticancer agent like Regorafenib (Re) benefited from a synergistic effect.

## 2. Materials and methods

### 2.1. Chemicals and reagents

The fetal colonic epithelial cell line FHC and colorectal cancer cell lines Colo205, HCT15, SW1116, NCI-H508, HT29, SW948, HCT116, and Caco-2, were acquired from ATCC (Manassas, VA, USA). Cell culture media including Dulbecco's modified Eagle's medium (DMEM), DMEM/F12, RPMI1640 (RPMI), Leibovitz's L-15 (L15), as well as supplements such as non-essential amino acids (NEAA), glutamax, fetal bovine serum (FBS), and penicillin/streptromycin (Pen/Strep) solution, were purchased from Gibco (USA). U-bottom ultra-low attachment 96-well plates (No. 7007) were obtained from Corning (NY, USA). N-Acetyl-D-Glucosamine (GlcNAc) was sourced from Phitsanuchemicals (Thailand). Dithiothreitol (DTT) was sourced from Amresco (USA).

Human epithelial growth factor (hEGF), transferrin, insulin, HEPES, OSMI-1 (Os), 3-(4,5-Dimethylthiazol-2-yl)-2,5-diphenyltetrazolium bromide (MTT), and dimethyl sulfoxide (DMSO) were procured from Sigma-Aldrich (USA). Annexin-FITC apoptosis reagent (A13199) and Propidium iodide (PI) were purchased from Thermofisher (Waltham, MA, USA). Irinotecan (Irinotel) was sourced from Fresenius Kabi Oncology (India). Cholera toxin and Regorafenib (Re) were obtained from MedChemExpress (USA).

Primary antibodies against O-GlcNAc (RL2, SC-59624, mouse monoclonal), OGA (SC-135093, rabbit monoclonal), and OGT (SC-32921, rabbit monoclonal) were acquired from Santa Cruz Biotechnology (Dallas, TX, USA). The primary antibody against actin (C4, MAB1501, mouse monoclonal) was obtained from Merck Millipore (Darmstadt, Germany). HRP-conjugated secondary antibodies, including goat-anti-rabbit and goat-anti-mouse IgG, were purchased from Thermofisher (Waltham, MA, USA) and Seracare (Milford, MA, USA), respectively. Polyvinylidene difluoride (PVDF) membranes were sourced from GE Healthcare (Chicago, IL, USA). X-ray film (Hyperfilm ECL) and chemiluminescent ECL reagents (Amersham ECL Prime, RPN2236) were procured from Cytiva (Marlborough, MA, USA).

### 2.2. Cell cultures

Colo205, HCT15 and NCI-H508 were cultured in RPMI supplemented with 10% FBS and 1% Pen/Strep. Caco-2 cells were cultured in DMEM supplemented with 10% FBS and 1% Pen/Strep. HCT116 and HT29 cells were cultured in DMEM supplemented with 10% FBS, 1% NEAA, 1% glutamax and 1% Pen/Strep. SW1116 and SW948 were cultured in L15 supplemented with 10% FBS and 1% Pen/Strep. FHC cells were cultured in DMEM/F12 supplemented with 10% FBS, 20 ng/mL hEGF, 5 μg/mL insulin, 5 μg/mL transferrin, 10 ng/mL cholera toxin, 100 ng/mL hydrocortisone, 50 μmol/L HEPES and 1% Pen/Strep. All cells were maintained under humidified atmosphere at 37˚C and 5% $CO_2$ except for SW1116 and SW948 which were cultured under atmospheric air.

### 2.3. Cytotoxicity tests

Cytotoxicity assessments were conducted in 96-well plates. Cells were initially seeded at a density of 8,000 cells per well and allowed to adhere overnight. For the cytotoxic evaluation of Os monotherapy, cells were exposed to varying concentrations of Os (100, 50, 20, 10, and 2 μM). Similarly, for the cytotoxic assessment of Re monotherapy, cells were treated with Re at

different concentrations (100 to 0.01 μM, with serial 10-fold dilutions). For the combination therapy of Os and Re, cells were treated with different concentrations of both agents. The non-cancerous cell line FHC served as the control for normal cells. DMSO, utilized for dissolving Os and Re, served as the vehicle control and was maintained at a final concentration of 0.5% (v/v) in the cell culture media for all experimental conditions. Following exposure to the test agents for 72 h, cell viability was determined using the MTT assay. All experiments were performed in triplicate. The results were expressed as the percentage of cell viability normalized to the vehicle control.

## 2.4. MTT assay

To assess cell viability, the MTT assay was conducted as previously published [23]. In brief, after reaching the designated exposure duration, the stock solution of MTT was added to a final concentration of 1.25 mg/mL and incubated for 4 h at 37°C. Then the culture media were aspirated, and DMSO was added at 100 μL per well. Absorbance at 550 nm was determined using a UV-Vis VICTOR Nivo Multimode Microplate Reader (PerkinElmer, Waltham, USA).

## 2.5. Western blot analysis

Western blot was performed as previously published [24], with some modifications. Cells were exposed to the designated test conditions and subsequently harvested. Cell lysis was performed using a lysis buffer supplemented with protease inhibitors, composed of 150 mM NaCl, 0.1% w/v sodium dodecyl sulfate, 0.5% w/v sodium deoxycholate, 1% v/v Triton NP-40, 50 mM Tris pH 8.0, along with 5 μg/mL leupeptin, 2 μg/mL aprotinin, 1 μg/mL pepstatin A, 1 mM PMSF, 1 mM benzamidine. 5 mM NaF, 1 mM sodium orthovanadate and 5 mM EDTA. Cell lysis was facilitated by sonication, and the resulting cell lysates were collected. The total protein concentration of each lysate was determined using the Bradford assay. SDS-PAGE electrophoresis was carried out at 160 V for 1 h. Subsequently, the resolved proteins were transferred onto PVDF membranes at 350 mA for 1.5 h. Membranes were then blocked with 5% nonfat dried milk in TBST for 1 h. Afterward, membranes were then probed with primary antibodies against O-GlcNAc (dilution 1:200), OGT (dilution 1:1000), OGA (dilution 1:1000), and actin (dilution 1:10000) overnight at 4°C. After four washes of 5 min each, the membranes were incubated with the appropriate secondary antibodies (anti-mouse IgG or anti-rabbit IgG) for 2 h and washed four times for 5 min each. Finally, membranes were treated with chemiluminescent ECL reagent and exposed to X-ray film for signal detection. The intensity of each band (for OGA, OGT, and actin) or lane (for O-GlcNAc) was quantified using the ImageJ program (National Institutes of Health, Bethesda, MD, USA), normalized to the actin loading control and further normalized to the sum of intensities across the nine cell lines to account for differences in exposure time (the sum of O-GlcNAc, OGT and OGA signal on each membrane was set to 100). Western blot experiments were performed in triplicate. Data were expressed as mean ± SD as shown in Fig 2C–2E.

## 2.6. Verification of antibody specificity for O-GlcNAc moieties

To increase intracellular O-GlcNAcylation, cells were treated with 1 mM DTT following the method described by Reeves et al. [25]. Briefly, HCT116 cells were exposed to 1 mM DTT for 4h. Cell pellet was then collected for western blot analysis, with untreated HCT116 cell lysates loaded in equal amount as a control.

To confirm that the RL2 antibody specifically reacts with GlcNAc moieties, HCT116 cell lysates were equally loaded for western blot analysis as described, transferred to PVDF membrane and cut into 4 pieces. One membrane was subjected to GlcNAc blocking as described by

Verathamjamras et.al. [26], by adding GlcNAc to a final concentration of 100 mM during the blocking and primary antibody incubation.

The control membrane was processed without GlcNAc addition. GlcNAc blocking was expected to reduce the reactivity of the O-GlcNAc-specific antibody.

Another membrane was subjected to on-blot β-elimination as outlined by Reeves et.al. [25], a method that selectively eliminated O-linked glycans while preserving N-linked glycans. The membrane was incubated with 55mM NaOH at 40˚C overnight, while the control membrane was incubated in $H_2O$ at the same temperature and duration. Afterward, membranes were washed three times in TBST, blocked with 5% nonfat dried milk in TBST and probed with antibodies as described in western blot protocol. For GlcNAc blocking and on-blot β-elimination, the treated membranes were placed side by side with the control membrane when ECL substrate was applied and then simultaneously exposed to X-ray film to ensure an equal exposure time.

## 2.7. Calculation and interpretation of pharmacological interaction by CompuSyn software

To assess the pharmacological interaction between Os and Re, the cell viability data from the combination experiment were analyzed using CompuSyn software with the Chou-Talalay method [27,28] as previously published [23]. The resulting combination index (CI) was employed to determine whether the interaction was synergistic (CI < 1), antagonistic (CI > 1), or additive (CI = 1). Additionally, the dose reduction index (DRI) was calculated, representing the fold reduction of drug dose in monotherapy relative to its dose in combined treatment required to achieve the same inhibitory effect. A DRI > 1 indicates favorability towards dose reduction.

## 2.8. Apoptosis analysis by flow cytometry

Flow cytometry examination of apoptotic cells was conducted as previously published [29] with some adjustments. In brief, NCI-H508 and FHC cells were subjected to test substances for 72 h. Subsequently, cells were harvested and resuspended in PBS for enumeration. For each replicate, $1 \times 10^5$ and $6.5 \times 10^4$ cells were collected for NCI-H508 and FHC cells, respectively, and stained with 5μl of FITC-conjugated annexin-V in binding buffer (0.1 M HEPES, 1.5 M NaCl, 50 mM KCl, 50 mM $MgCl_2$, 18 mM $CaCl_2$, pH 7.4) for 15 min under light protection at room temperature. Then, propidium iodide was added to reach 5 μg/mL final concentration and incubated for 15 min under light protection at room temperature. Cells were then analyzed with a flow cytometer (Attune NxT, Thermo Fischer). Data analysis was performed using Attune NxT Software (Thermo Fischer). All experiments were conducted in triplicate.

## 2.9. Cell cycle distribution analysis

Flow cytometry analysis of cell cycle distribution by propidium iodide staining was conducted as previously published [29] with some adjustments. In brief, NCI-H508 cells were subjected to test agents for 72 h. Subsequently, cells were washed with PBS, harvested and fixed with 70% ice-cold ethanol. Subsequently, cells were washed with ice-cold PBS twice and treated for 5 min at room temperature with 100 μg/mL of DNase free RNase A in PBS containing 0.1% v/v Triton-X 100. Alterward, cells were stained with 20 μg/mL propidium iodide in PBS containing 0.1% v/v Triton-X 100 while protected from light for 15 min at room temperature. Cells were then analyzed with a flow cytometer (Attune NxT, Thermo Fischer). Data analysis was performed using Attune NxT Software (Thermo Fischer). All experiments were conducted in triplicate.

## 2.10. 3D spheroid culture

Spheroid culture were performed in 96-well U-bottom ultralow adherent plates. NCI-H508 cells were seeded at a density of 300 cells/well and incubated for 2 d to allow spheroid formation. Afterward, spheroids were exposed to Os, Re or their combinations for 9 d (8 spheroids/ treatment condition). Images of the spheroids were captured under 40x magnification using an inverted microscope (Eclipse TE 2000-U, Nikon, Japan). To quantify spheroid size, the images were analyzed to measure their areas using image J software.

## 2.11. Data analysis

All data were shown as means ± standard deviations (SD). One-way ANOVA with Tukey's HSD post-hoc test was used to determine the statistical significance between treatment multiple groups (Figs 1, 2 and 4–7). Two-tailed Student's t-test was used to determine the statistical significance between two groups (box & whisker plots in Figs 2 and S3). Pearson correlation coefficient ($r$) was calculated to determine the correlation between O-GlcNAc-related signatures (i.e. total O-GlcNAcylation, OGT, and OGA protein expression levels) and sensitivity to Os using Microsoft Excel 2016 (Fig 3). Half-maximal inhibitory concentration ($IC_{50}$) was computed using the non-linear regression log (inhibitor) vs. normalized response - Variable slope [30]. GraphPad Prism software version 7 was used for $IC_{50}$ calculations and statistical analysis. $p$-value $< 0.05$ signified statistical significance.

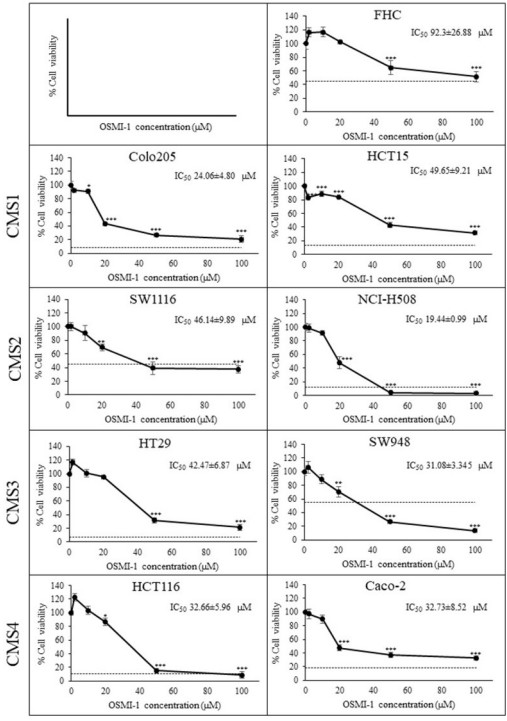

**Fig 1. Cytotoxic activity and $IC_{50}$ of Os against CRC cell lines from different molecular subtypes.** FHC cells were used as a non-cancerous cell control. The dashed line (---) represented % cell viability following treatment with 100 µM Irinotecan (Iri) as the positive control. Data presented as mean ± SD (n = 3). *, **, *** $p < 0.05$, 0.01 and 0.001 compared to the vehicle control, respectively.

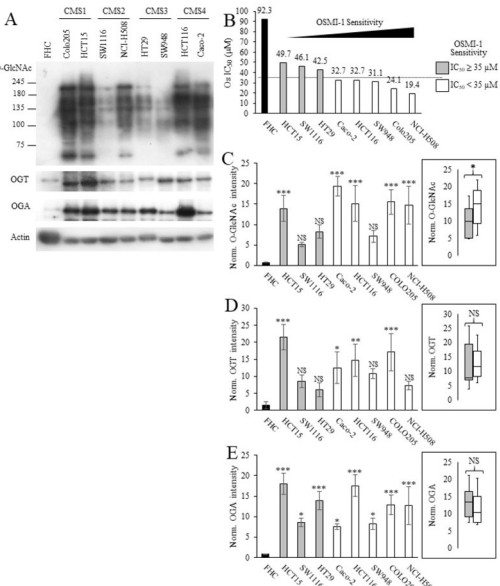

**Fig 2. The level of OGT, OGA and O-GlcNAcylation of intracellular proteins in CRC cell lines from different CMS molecular subclass.** (A) Representative western blot images of protein O-GlcNAcylation, OGT and OGA in CRC cell lines. (B) CRC cell lines sorted by Os sensitivity from low to high (left to right). The dashed line (---) represented the mean Os IC$_{50}$ (~35 μM). Gray bars represented CRC cells having Os IC$_{50} \geq 35$ μM. White bars represented CRC cells having Os IC$_{50} < 35$ μM. Normalized (C) O-GlcNAc, (D) OGA, and (E) OGT intensities of each cell line (bar graph) or CRC cell lines categorized into two groups based on Os IC$_{50}$ (box & whisker plot). Data presented as mean ± SD (n = 3), *, **, *** $p < 0.05$, 0.01 and 0.001, respectively.

## 3. Results

### 3.1. Cytotoxicity profile of Os against CRC cell lines

Firstly, to evaluate the toxicity profile of Os against CRC cells, eight CRC cell lines, comprising two cell lines from each consensus molecular subtypes (CMSs) [31], were subjected to Os treatment at various concentrations (Fig 1). Indeed, Os treatment elicited a dose-dependent suppression of cell viability across all CRC cell lines, with IC$_{50}$ values ranging from approximately 20 to 50 μM. Notably, the toxicity of Os did not exhibit discernible specificity towards any particular CMS subclass. Moreover, the cytotoxic effects of Os appeared to be more pronounced in CRC cells compared to normal colonic epithelial cells, as evidenced by the response of the FHC cell line (IC$_{50}$ ~ 90 μM), suggesting the presence of a therapeutic window exploitable for CRC treatment. Among the CRC cell lines examined, NCI-H508 emerged as the most susceptible to Os, as indicated by the lowest IC$_{50}$ value.

### 3.2. The correlation between cellular O-GlcNAcylation and OS sensitivity

Subsequently, we asked whether the toxicity of Os correlated with the O-GlcNAcylation of intracellular proteins or the expression levels of the two governing enzymes, OGT and OGA. We thus conducted western blot analysis using specific antibodies against O-GlcNAc, OGT, and OGA in triplicate (S1A–S1C Fig). Note that S1A Fig was selected as a representative image shown in Fig 2A. Our results revealed a notable increase in global O-GlcNAcylation, as well as OGT and OGA protein expression, in CRC cells compared to the normal cell control FHC. To confirm that the antibody specifically detected O-GlcNAc, we conducted western blot using HCT116 cell lysates under various stress conditions, including treatment with 1 mM DTT, blocking with 100 mM GlcNAc, and on-blot β-elimination which removed O-linked glycans

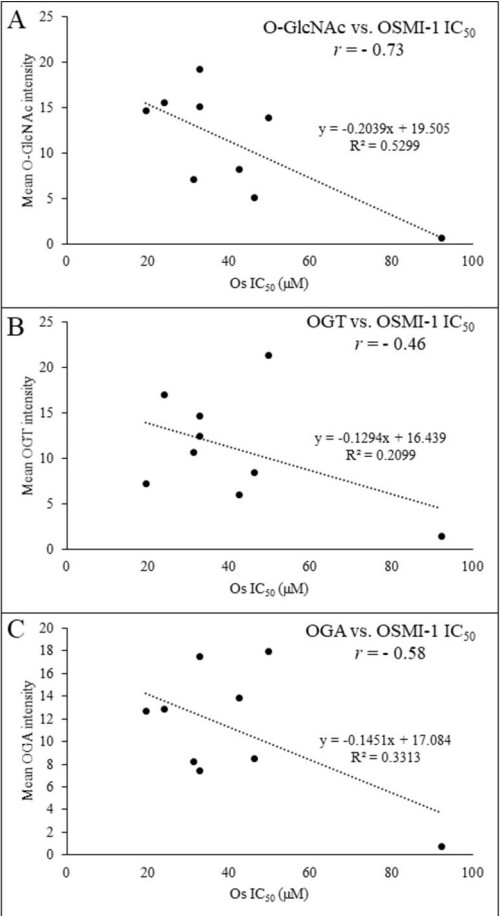

**Fig 3. The correlation between O-GlcNAcylation pathway and sensitivity to OGTi in CRC cell lines.** Scatter plot between $IC_{50}$ of Os vs. mean (A) O-GlcNAcylation, (B) OGT protein expression and (C) OGA protein expression. $r$, Pearson correlation coefficient.

(S2 Fig) [25,26]. As reported by other research groups, 1 mM DTT treatment increased total protein O-GlcNAcylation (S2 Fig, lane 2 vs. lane 1 control). In contrast, the O-GlcNAc signal was markedly reduced by the presence of 100 mM GlcNAc in the blocking buffer used with the

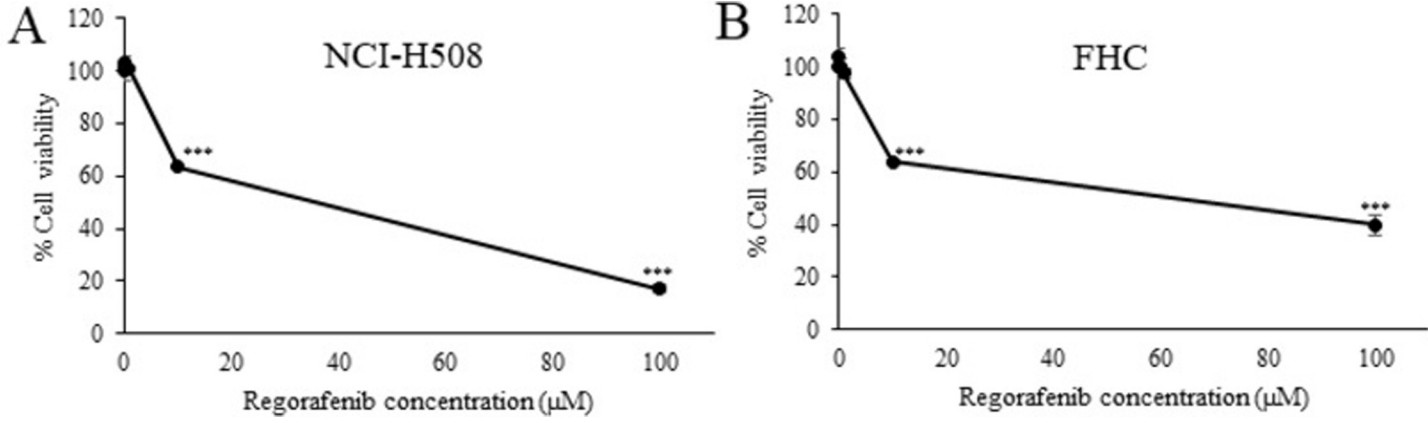

**Fig 4. Cytotoxic activity of Re against NCI-H508 and FHC cells.** Data presented as mean ± SD (n = 3). *** $p < 0.001$ compared to the vehicle control.

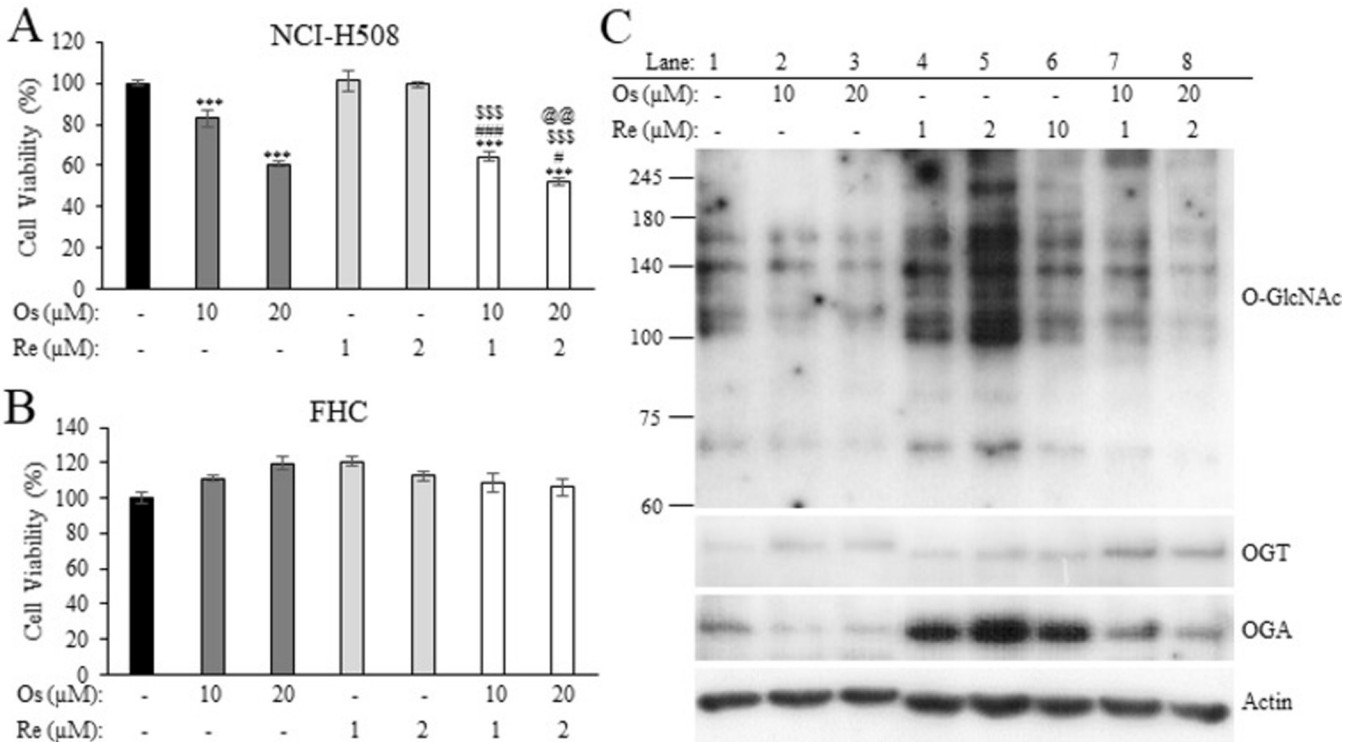

**Fig 5.** Cytotoxic activity of Os and Re combination at a molar ratio 10:1 (Os:Re) against NCI-H508 (A) and FHC (B) cells determined by MTT assay. Data presented as mean ± SD (n = 3). *** $p < 0.001$ compared to the vehicle control. #, ## $p < 0.05$ and 0.01 compared to Os monotherapy at the same concentration, respectively. \$ \$ \$ \$ $p < 0.001$ compared to Re monotherapy at the same concentration. @@ $p < 0.01$ compared to Os:Re 10:1 combination therapy. (C) Changes in NCI-H508 total O-GlcNAcylation, OGT and OGA protein expression levels upon exposure of Os, Re or their combinations determined by western blot.

anti-O-GlcNAc antibody (S2 Fig, lane 4 vs. lane 3 control). A similar reduction was observed in the membrane subjected to on-blot β-elimination (S2 Fig, lane 6 vs. lane 5 control).

We then sorted the CRC cells based on their Os $IC_{50}$ values (Fig 2B) from high to low and utilized the mean Os $IC_{50}$ (~35 μM) as the cut-off to divide the cells to two groups: those with high sensitivity (white bars) and those with low sensitivity (gray bars) to Os. The intensity of O-GlcNAc, OGT, and OGA signals in each CRC cell was quantified and presented in the same order (Fig 2C–2E, respectively). Intriguingly, CRC cells exhibiting high sensitivity to Os displayed significantly elevated levels of intracellular O-GlcNAc (Fig 2C, box & whisker plot, $p = 0.0154$) compared to those less sensitive to Os. However, no discernible trend was observed in OGT and OGA levels between the two groups (Fig 2D–2E). Furthermore, we examined the correlation between O-GlcNAc-related signatures and OGTi sensitivity through a scatter plot analysis to calculate the Pearson correlation coefficient ($r$, Fig 3). Remarkably, a strong negative correlation ($r = -0.73$) was evident between Os sensitivity and total protein O-GlcNAcylation (Fig 3A), while moderate correlations were observed between Os sensitivity versus OGT and OGA expression ($r = -0.46$ and $-0.58$, respectively) (Fig 3B and 3C). Taken together, our data suggest that intracellular O-GlcNAc levels may serve as a more robust marker than OGT or OGA protein expression in predicting Os sensitivity in CRC cells.

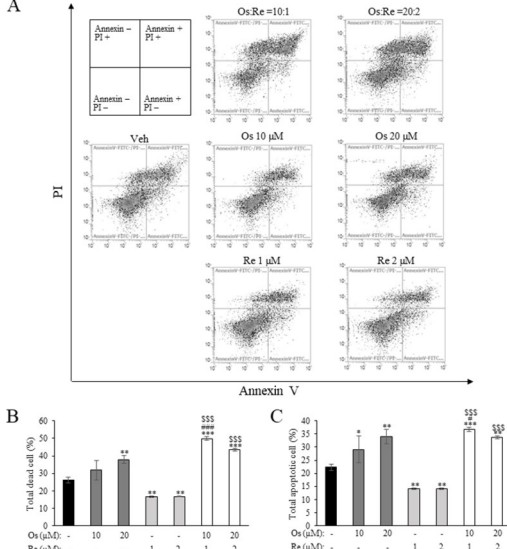

**Fig 6. The effect of Os and Re combination at a molar ratio 10:1 (Os: Re) on NCI-H508 cell death.** (A) Cellular apoptosis/necrosis determined by Annexin-V/PI double staining and flow-cytometry analysis. The bar graph depicted (B) total dead cells and (C) total apoptotic cells. Data presented as mean ± SD, n = 3. **, *** $p < 0.01$ and 0.001 compared to the vehicle control, respectively. #, ### $p < 0.05$ and 0.001 compared to Os monotherapy at the same concentration, respectively. $ $ $ $p < 0.001$ compared to Re monotherapy at the same concentration, respectively.

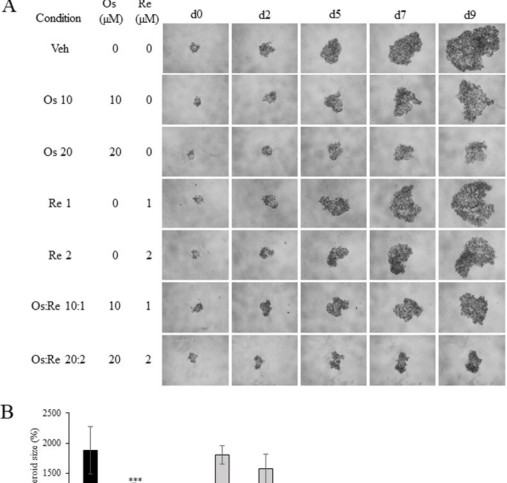

**Fig 7. The effect of Os and Re combination at molar ratio 10:1 (Os: Re) on the growth of NCI-H508 spheroids.** (A) Representative images of spheroids exposed to Os, Re or their combinations at depicted concentrations from 0 to 9 days after treatment. (B) Normalized spheroid size at 9 days after treatment. Data presented as mean ± SD (n = 8). *** $p < 0.001$ compared to vehicle control. # $p < 0.05$ compared to Os monotherapy at the same concentration. $ $ $ $p < 0.001$ compared to Re monotherapy at the same concentration. @ $p < 0.05$ compared to Os:Re 10:1 combination therapy.

### 3.3. Synergistic anticancer activity between Os and Re

Since the majority of anticancer therapies rely on the use of multiple drugs to leverage pharmacological synergy, we investigated whether Os could be employed in combination with another anticancer agent to achieve enhanced inhibitory effects. For this purpose, we selected the multi-kinase inhibitor regorafenib (Re) as a model drug. First, cytotoxic activity of Re was examined in NCI-H508 and FHC cells (Fig 4). Indeed, a dose-dependent cytotoxicity was observed at concentration higher than 1 μM. To explore the interaction between Os and Re, NCI-H508 cells were treated with varying concentrations of both agents. Specifically, cells were exposed to Os at 10, 15, 20, and 30 μM or Re at 1, 5, 10, and 20 μM, either individually or simultaneously, utilizing a checkerboard design (Table 1). Subsequently, the cell viability data obtained was analyzed using Compusyn software to calculate the combination index (CI) and determine the pharmacological interaction—whether antagonistic (CI >1), additive (CI = 1), or synergistic (CI <1)—using the Chou-Talalay method [27,28] (Table 2). The results unveiled a synergistic interaction between Os and Re at certain combinations, particularly those involving Os concentrations ranging from 10 to 20 μM and Re at 1 μM. Given that the combination containing Os and Re at 10 and 1 μM (Os:Re 10:1), respectively, yielded the lowest CI value, this molar ratio was selected for further investigation.

### 3.4. The synergism between Os and Re is partly explained by Os-mediated reversal of increased cellular O-GlcNAcylation induced by Re treatment

Based on the promising cytotoxic activity observed with the combination Os:Re 10:1, we proceeded to investigate this combination ratio in NCI-H508 cells (Fig 5A) and the normal colonic cell line FHC (Fig 5B). Interestingly, the cytotoxic activity of the Os:Re 10:1 and 20:2 dual treatments against NCI-H508 cells was significantly higher than that of monotherapy at equivalent concentrations, while exhibiting minimal cytotoxicity against non-cancerous cells (Fig 5B). Concurrently, we explored the levels of intracellular O-GlcNAcylation and the protein expression of OGT and OGA upon exposure to Os, Re, or their combined treatment using western blot analysis (Fig 5C). As anticipated, a reduction in O-GlcNAcylation was observed in NCI-H508 cells treated with Os (lanes 2–3). Consistent with previous reports by independent research groups, Os treatment led to an increase in OGT and a decrease in OGA protein levels, indicating a cellular compensatory mechanism to maintain intracellular O-GlcNAcylation levels upon OGT inhibition [32,33]. Interestingly, treatment of cells with Re resulted in a marked increase in cellular O-GlcNAcylation especially at low concentrations (lane 4–5), which was reversed upon addition of Os (lane 7–8). Furthermore, it is noteworthy that while the Re-mediated increase in O-GlcNAcylated proteins was accompanied by a robust increase

**Table 1. The effect of Os, Re and Os-Re combination on the viability of NCI-H508 cells.**

| | | Regorafenib (μM) | | | | | | | | | |
| | | 0 | | 1 | | 5 | | 10 | | 20 | |
| | | Mean | SD | Mean | SD | Mean | SD | Mean | SD | Mean | SD |
|---|---|---|---|---|---|---|---|---|---|---|---|
| OSMI-1 (μM) | 0 | 100.00 | 2.44 | 85.86 | 7.82 | 40.75 | 3.09 | 32.35 | 4.71 | 27.25 | 4.04 |
| | 10 | 109.09 | 1.66 | 52.86 | 6.02 | 53.71 | 0.77 | 38.97 | 3.27 | 28.84 | 2.85 |
| | 15 | 86.42 | 1.23 | 45.44 | 3.97 | 46.04 | 1.30 | 32.96 | 2.09 | 19.99 | 3.18 |
| | 20 | 38.60 | 3.47 | 33.39 | 4.26 | 38.78 | 2.69 | 16.78 | 1.75 | 14.01 | 1.88 |
| | 30 | 24.66 | 0.43 | 13.11 | 0.69 | 16.11 | 0.52 | 8.75 | 2.27 | 4.70 | 0.27 |

Data presented as mean and SD (n = 3).

**Table 2. The interaction of Os and Re, and DRI determined by CompuSyn software.**

| Os (μM) | Re (μM) | Molar Ratio Os:Re | % Viability | % Inhibition (100-% Viability) | CI | Interaction | DRI Os | DRI Re |
|---|---|---|---|---|---|---|---|---|
| 0 | 20 | 0.5 | 28.84 | 71 | 1.9 | antagonism | 2.66 | 0.66 |
| 15 | 20 | 0.75 | 19.99 | 80 | 1.42 | antagonism | 2.00 | 1.09 |
| 10 | 10 | 1 | 38.97 | 61 | 1.63 | antagonism | 2.38 | 0.83 |
| 20 | 20 | 1 | 14.01 | 86 | 1.19 | antagonism | 1.67 | 1.69 |
| 15 | 10 | 1.5 | 32.96 | 67 | 1.52 | antagonism | 1.69 | 1.08 |
| 30 | 20 | 1.5 | 4.7 | 95 | 0.86 | synergism | 1.47 | 5.42 |
| 10 | 5 | 2 | 53.71 | 46 | 1.62 | antagonism | 2.04 | 0.88 |
| 20 | 10 | 2 | 16.78 | 83 | 1.01 | additive | 1.58 | 2.67 |
| 15 | 5 | 3 | 46.04 | 54 | 1.49 | antagonism | 1.48 | 1.23 |
| 30 | 10 | 3 | 8.75 | 91 | 0.97 | additive | 1.26 | 5.66 |
| 20 | 5 | 4 | 38.78 | 61 | 1.45 | antagonism | 1.19 | 1.65 |
| 30 | 5 | 6 | 16.11 | 84 | 1.11 | antagonism | 1.07 | 5.76 |
| 10 | 1 | 10 | 52.86 | 47 | 0.7 | synergism | 2.06 | 4.60 |
| 15 | 1 | 15 | 45.44 | 55 | 0.83 | synergism | 1.49 | 6.41 |
| 20 | 1 | 20 | 33.39 | 67 | 0.88 | synergism | 1.27 | 10.81 |
| 0 | 1 | 30 | 13.11 | 87 | 0.91 | additive | 1.14 | 36.99 |

The combinations exhibiting synergistic relationship were highlighted. CI = Combination index, DRI = Dose reduction index.

in OGA expression, OGT protein levels remained largely unchanged. Therefore, we hypothesized that Re treatment increased intracellular O-GlcNAcylation possibly by enhancing OGT enzymatic activity, which was counteracted by Os, thus explaining the observed synergism between the two agents at low Re concentrations (1–2 μM). Nonetheless, further experiments are warranted to validate the effect of Re on OGT and OGA enzymatic activity.

## 3.5. Combined Os:Re treatment increased NCI-H508 cell death

Considering the promising cytotoxic activity observed with Os:Re 10:1 and 20:2, we sought to determine whether this effect was attributable to increased cell death. To this end, we conducted flow cytometry analysis on NCI-H508 cells treated with Os:Re 10:1, 20:2, and monotherapy at corresponding concentrations, utilizing FITC-conjugated Annexin-V and PI staining (Fig 6A). Consistent with the cell viability data, our results revealed a significant increase in total dead cells (comprising Annexin+/PI-, Annexin-/PI+, and Annexin+/PI+ populations) and total apoptotic cells (comprising Annexin+/PI- and Annexin+/PI+ populations) in the Os:Re 10:1 treatment group compared to monotherapy (Fig 6B and 6C). A similar trend was observed with Os:Re 20:2 treatment, although the increase in dead and apoptotic cells was not statistically significant compared to Os monotherapy. These findings collectively suggested that the cytotoxic activity of the dual therapy was associated with elevated cell death, at least under the conditions of concentration and exposure time investigated (72 h). Next, to confirm that the combination therapy elicited limited cytotoxicity against non-cancerous cells, flow cytometry was performed on FHC cells treated with Os:Re 20:2 combination (S3A Fig). The result revealed minimal change in the percentage of dead and apoptotic cells in the treatment group compared to that of the vehicle control (S3B and S3C Fig, respectively), consistent with cell viability data (Fig 5A). Furthermore, it might be noteworthy to mention that cell cycle distribution of NCI-H508 cells remained largely unaffected by Os and Re treatment at these concentrations (S4 Fig), suggesting that the elevated cell death was likely cell-cycle independent.

### 3.6. Combined Os:Re treatment inhibited NCI-H508 spheroid growth

Finally, the anticancer efficacy of the Os:Re combination treatment was evaluated against NCI-H508 3D spheroids (Fig 7). Indeed, Os treatment led to a suppression of spheroid growth, whereas Re treatment resulted in minimal inhibition (Fig 7A), indicating that the cytotoxic effects of Os and Re observed in 2D cell culture were recapitulated in our 3D spheroid model. Interestingly, the Os:Re 10:1 combination treatment exhibited significantly enhanced inhibitory activity compared to monotherapy at equivalent concentrations (Fig 7B). Treatment with Os:Re at a ratio of 20:2 demonstrated a notably higher inhibitory effect compared to the 10:1 condition and Re monotherapy at 2 μM; however, such increase was not statistically significant when compared to Os monotherapy at 20 μM.

## 4. Discussion

To our knowledge, there have been no previous reports directly examining the OGTi-mediated decrease in the viability of colorectal cancer cells. However, Lee et al. reported that Os treatment of HCT116 cells was accompanied by an increased cytochrome C [34], suggesting that Os treatment is associated with increased cancer cell death. *In vivo* experiments with mice revealed that Os treatment at 1 mg/kg IV once per day for 3 weeks decreased tumor volume in HCT116 tumor xenografts [34]. Additionally, OGTi treatment in other cancer types, including prostate, breast, endometrial, liver, pancreatic, and bladder cancer, elicited beneficial anticancer effects, such as suppression of cell viability, cell growth, migration, and invasion [12,33,35–40]. Similar effects were observed in cancer cells treated with si/shRNA targeting OGT [41–43]. Moreover, the anticancer activity of OGTi is likely a class effect, as similar activities were reported with other OGTi compounds such as OSMI-2 (16) and ST045849 (33). We also reported that OGTi exerted a synergistic anticancer activity with the multi-kinase inhibitor regorafenib. In line with our data, other research groups also reported the synergism of OGTi with other anticancer agents [36,37]. Finally, OGTi was tolerable in animal, at least to a certain extent, as Os dose as high as 10 mg/kg administered intravenously has been investigated in mice by several independent research groups [34,38,40,44]. Together, these data supported the value and feasibility of OGTi as a potential anticancer agent for various human malignancies both as mono and combined therapy. Further researches are thus warranted to assess the activity of OGTi in various cancer types and perhaps to assess whether the addition of OGTi to established anticancer agents or regimes confers therapeutic benefits.

Although sensitivity to OGTi did not exhibit any detectable correlation with a specific CMS molecular subclass, our data provided evidence that OGTi exerted inhibitory activities against colorectal cancer cell lines and could be used synergistically with certain anticancer agents, such as regorafenib. Moreover, cellular O-GlcNAcylation was identified as a better predictive marker for OGTi sensitivity than the expression levels of the two O-GlcNAc-regulating proteins, namely OGT and OGA. Further research efforts should thus focus on the development of rapid and high-throughput assays to measure intracellular O-GlcNAcylation which will pave the way for establishing reliable criteria to pinpoint OGTi-sensitive cancers.

## Supporting information

**S1 Raw images. Raw images underlying all western blot results.**
(PDF)

**S1 Fig. Western blot data showing OGT, OGA and total O-GlcNAcylation levels of CRC cell lines from different CMS molecular subclass performed in triplicate (A-C).** FHC cells

were used as a non-cancerous cell control.
(TIF)

**S2 Fig. The anti-O-GlcNAc antibody (RL2) specifically reacted to O-GlcNAc.** Western blot analysis of HCT116 cell lysates under various conditions: Lane 1, 3 and 5, untreated HCT116 cell lysate controls; lane 2, HCT116 treated with 1 mM DTT for 4h; lane 4, 100 mM GlcNAc blocking; and lane 6, on-blot β-elimination. Note that lane 1 and 2 were on the same blot, while lane 3–6 were duplicated blots subjected to different processing, as described in materials and methods.
(TIF)

**S3 Fig. The effect of Os and Re combination at a molar ratio 20:2 (Os:Re) on FHC cell death.** (A) Cellular apoptosis/necrosis determined by Annexin-V/PI double staining and flow-cytometry analysis. The bar graph depicted (B) total dead cells and (C) total apoptotic cells. Data presented as mean ± SD, n = 3. NS, $p > 0.05$.
(TIF)

**S4 Fig. The effect of Os and Re combination at molar ratio 10:1 (Os:Re) on cell cycle distribution of NCI-H508.** (A) Representative histogram plot of cells stained with PI analyzed by flow-cytometry. (B) Quantitation of cells in G0/G1, S and G2/M phases of the cell cycle based on PI staining status. Data presented as mean ± SD (n = 3).
(TIF)

## Acknowledgments

We gratefully acknowledged the Faculty of Pharmacy, Silpakorn University, for providing the essential facilities and research equipment necessary for the completion of this study.

## Author Contributions

**Conceptualization:** Pawaris Wongprayoon, Purin Charoensuksai.

**Formal analysis:** Pawaris Wongprayoon, Supusson Pengnam, Purin Charoensuksai.

**Funding acquisition:** Purin Charoensuksai.

**Investigation:** Pawaris Wongprayoon, Supusson Pengnam, Roongtiwa Srisuphan, Purin Charoensuksai.

**Methodology:** Pawaris Wongprayoon, Supusson Pengnam, Purin Charoensuksai.

**Project administration:** Purin Charoensuksai.

**Resources:** Pawaris Wongprayoon, Praneet Opanasopit, Siwanon Jirawatnotai, Purin Charoensuksai.

**Supervision:** Pawaris Wongprayoon, Praneet Opanasopit, Siwanon Jirawatnotai, Purin Charoensuksai.

**Validation:** Purin Charoensuksai.

**Visualization:** Pawaris Wongprayoon, Supusson Pengnam, Siwanon Jirawatnotai, Purin Charoensuksai.

**Writing – original draft:** Pawaris Wongprayoon, Supusson Pengnam, Purin Charoensuksai.

**Writing – review & editing:** Pawaris Wongprayoon, Supusson Pengnam, Praneet Opanasopit, Siwanon Jirawatnotai, Purin Charoensuksai.

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
