## [Decision Letter · Decision Letter 0]

16 Aug 2024

PONE-D-24-28155The CorrelationBetween Cellular O-GlcNAcylation and Sensitivity to O-GlcNAc Inhibitor in Colorectal Cancer CellsPLOS ONE

Dear Dr. Charoensuksai,

Thank you for submitting your manuscript to PLOS ONE. After careful consideration, we feel that it has merit but does not fully meet PLOS ONE’s publication criteria as it currently stands. Therefore, we invite you to submit a revised version of the manuscript that addresses the points raised during the review process.

We look forward to receiving your revised manuscript.

Kind regards,

Sadiq Umar

Academic Editor

PLOS ONE

Journal Requirements:

This report is the product of the research project "The effect of OGT inhibitor in the four molecular subclasses of colorectal cancer," financially supported by the Research Grant for New Scholar (RGNS) from the Office of the Permanent Secretary, Ministry of Higher Education, Science, Research, and Innovation, Fiscal Year 2563BE [Grant number RGNS 63-225]. 

This report is the product of the research project "The effect of OGT inhibitor in the four molecular subclasses of colorectal cancer," financially supported by the Research Grant for New Scholar (RGNS) from the Office of the Permanent Secretary, Ministry of Higher Education, Science, Research, and Innovation, Fiscal Year 2563BE [Grant number RGNS 63-225]. 

This report is the product of the research project "The effect of OGT inhibitor in the four molecular subclasses of colorectal cancer," financially supported by the Research Grant for New Scholar (RGNS) from the Office of the Permanent Secretary, Ministry of Higher Education, Science, Research, and Innovation, Fiscal Year 2563BE [Grant number RGNS 63-225]. 

In your cover letter, please note whether your blot/gel image data are in Supporting Information or posted at a public data repository, provide the repository URL if relevant, and provide specific details as to which raw blot/gel images, if any, are not available. Email us at plosone@plos.org if you have any question.

Reviewers' comments:

Reviewer's Responses to Questions

**Comments to the Author**

1. Is the manuscript technically sound, and do the data support the conclusions?

Reviewer #1: Partly

Reviewer #2: Yes

2. Has the statistical analysis been performed appropriately and rigorously? 

Reviewer #1: Yes

Reviewer #2: Yes

3. Have the authors made all data underlying the findings in their manuscript fully available?

Reviewer #1: Yes

Reviewer #2: Yes

4. Is the manuscript presented in an intelligible fashion and written in standard English?

Reviewer #1: Yes

Reviewer #2: Yes

5. Review Comments to the Author

**Reviewer #1: **Charoensuksai et al, studied the relation O-GlcNAcylation with colorectal cancer. They measured O-GlcNAcylation, OGA and OGT levels levels in colorectal cancer cell lines and their sensitivity to OGT inhibitor OSML1. They found a dose dependent decline in cell viability in cancer cell lines and Os exhibited a synergistic relationship with regorafenib (Re). They concluded that OSML1 can be a potent anticancer agent when used along with other treatments. It is a well thought study, but the data supporting the claims especially the western blots are low quality. The authors should use proper controls to check the specificity of antibodies especially when measuring O-GlcNAc. Stressing Cells to enhance O-GlcNAcylation can be a good control. I would suggest using Lectins as well for secondary confirmation. Inclusion of one more control line is highly recommended.

**Reviewer #2:** The manuscript suggests a novel predictive marker, O-GlcNAc transferase inhibitor (OGTi), for O-GlcNAcylation in various cancer cell lines, particularly colorectal cell lines. The manuscript is acceptable after addressing minor grammatical errors and the following comments.

Comments and Suggestions

Background on Regorafenib (Re): The authors should provide a brief background on regorafenib, explaining why it was chosen for the study. Additionally, they should clarify its FDA approval status.

Error Bars and Statistical Analysis: In Figure 2 C, D, and E, the western blot quantification lacks error bars for the intensity of O-GlcNAc, OGA, and OGT in all cell lines. Given that the experiments were performed in triplicate, the authors should calculate the data from all three results, include error bars, and conduct statistical analysis.

Grammatical Error: In paragraph 3.2, "our results evealed..." should be corrected to "our results revealed...".

Flow Cytometry Data: While Figure 6 demonstrates the cytotoxic activity of the combination of Os:Re in NCI-H508 cells, suggesting the efficacy of dual therapy, flow cytometry data is missing for FHC cells (control cells). Including similar flow cytometry experiments for FHC cells would provide valuable insights into the specificity of the treatment.

6. PLOS authors have the option to publish the peer review history of their article (what does this mean?). If published, this will include your full peer review and any attached files.

Reviewer #1: **Yes: **Willayat Yousuf Wani

Reviewer #2: **Yes: **Mohd Tayyab

---

## [Author Response · Author response to Decision Letter 0]

25 Sep 2024

Reviewer 1's comment:

Charoensuksai et al, studied the relation O-GlcNAcylation with colorectal cancer. They measured O-GlcNAcylation, OGA and OGT levels levels in colorectal cancer cell lines and their sensitivity to OGT inhibitor OSML1. They found a dose dependent decline in cell viability in cancer cell lines and Os exhibited a synergistic relationship with regorafenib (Re). They concluded that OSML1 can be a potent anticancer agent when used along with other treatments. It is a well thought study, but the data supporting the claims especially the western blots are low quality. The authors should use proper controls to check the specificity of antibodies especially when measuring O-GlcNAc. Stressing Cells to enhance O-GlcNAcylation can be a good control. I would suggest using Lectins as well for secondary confirmation. Inclusion of one more control line is highly recommended.

Response: 

We appreciate the reviewer’s suggestion and have added three additional tests, previously reported by other research groups, to verify antibody specificity to O-GlcNAc moieties: (1) stressing cells with DTT to increase O-GlcNAcylation, (2) adding free GlcNAc to the blocking solutions during the blocking and antibody incubation process (GlcNAc blocking), and (3) performing on-blot β-elimination to remove O-glycan under basic conditions.

The results, presented in the new S2 Fig, confirm the specific detection of O-GlcNAc modifications on proteins and are described in the revised manuscript on page 11-12, lines 264-271. The sources of materials, including GlcNAc and DTT, have been added to the “Chemicals and Reagents” section on page 5, lines 101-102. The detailed procedure is outlined in section “2.6. Verification of antibody specificity for O-GlcNAc moieties” on page 8, lines 170-187.

References to the employed protocols include:

25. Reeves RA, Lee A, Henry R, Zachara NE. Characterization of the specificity of O-GlcNAc reactive antibodies under conditions of starvation and stress. Anal Biochem. 2014;457:8-18. doi: 10.1016/j.ab.2014.04.008.

26. Verathamjamras C, Sriwitool TE, Netsirisawan P, Chaiyawat P, Chokchaichamnankit D, Prasongsook N, et al. Aberrant RL2 O-GlcNAc antibody reactivity against serum-IgA1 of patients with colorectal cancer. Glycoconj J. 2021;38(1):55-65. doi: 10.1007/s10719-021-09978-8.

Reviewer 2's comments:

1. Background on Regorafenib (Re): The authors should provide a brief background on regorafenib, explaining why it was chosen for the study. Additionally, they should clarify its FDA approval status.

Response: 

The background of regorafenib, including its clinical approval status, was added in the introduction section page 3 line 56 - 70 as per the reviewer’s suggestion. 

2. Error Bars and Statistical Analysis: In Figure 2 C, D, and E, the western blot quantification lacks error bars for the intensity of O-GlcNAc, OGA, and OGT in all cell lines. Given that the experiments were performed in triplicate, the authors should calculate the data from all three results, include error bars, and conduct statistical analysis.

Response:

We appreciate the reviewer’s comment and have responded by performing western blotting in triplicate. Data from all three replicates were analyzed, and the graphs have been adjusted to include error bars. Statistical analysis was also conducted, as recommended. Fig 2 and 3 have been revised accordingly, and a new S1 Fig has been added to display the western blot signals for O-GlcNAc, OGT, and OGA from the three replicates. The information related to western blot under “Materials and methods” and “Results” sections have been updated to reflect these changes.

3. Grammatical Error: In paragraph 3.2, "our results evealed..." should be corrected to "our results revealed...".

Response:

Corrected as per the reviewer’s suggestion.

4. Flow Cytometry Data: While Figure 6 demonstrates the cytotoxic activity of the combination of Os:Re in NCI-H508 cells, suggesting the efficacy of dual therapy, flow cytometry data is missing for FHC cells (control cells). Including similar flow cytometry experiments for FHC cells would provide valuable insights into the specificity of the treatment.

Response:

We appreciate the reviewer’s suggestion and have conducted flow cytometry analysis on FHC cells. However, due to their non-cancerous nature, FHC cells grow significantly slower than cancerous cell lines, which presented a challenge in expanding them to yield sufficient quantities for flow cytometry at all concentration intervals and combinations, as we did with NCI-H508 cells. Therefore, we selected the most extreme condition, the Os:Re 20:2 combination, for flow cytometry analysis of FHC cells compared to the vehicle control. The results, shown in the new S3 Fig, revealed limited changes in the total apoptotic and dead cells, which aligned with the MTT assay cell viability data (Fig. 5A). This new finding is now described on page 17, lines 368-372, and the experimental procedure has been updated to include this experiment on page 9, lines 202-203.

---

## [Editor Report · Decision Letter 1]

3 Oct 2024

The Correlation Between Cellular O-GlcNAcylation and Sensitivity to O-GlcNAc Inhibitor in Colorectal Cancer Cells

PONE-D-24-28155R1

Dear Dr. Charoensukai,

We’re pleased to inform you that your manuscript has been judged scientifically suitable for publication and will be formally accepted for publication once it meets all outstanding technical requirements.

Kind regards,

Sadiq Umar

Academic Editor

PLOS ONE

---

## [Editor Report · Acceptance letter]

7 Oct 2024

PONE-D-24-28155R1 

PLOS ONE

Dear Dr. Charoensuksai, 

I'm pleased to inform you that your manuscript has been deemed suitable for publication in PLOS ONE. Congratulations! Your manuscript is now being handed over to our production team.

Kind regards, 

on behalf of

Dr. Sadiq Umar 

Academic Editor

PLOS ONE